# Current Insights into the Maturation of Epstein–Barr Virus Particles

**DOI:** 10.3390/microorganisms12040806

**Published:** 2024-04-17

**Authors:** Asuka Nanbo

**Affiliations:** National Research Center for the Control and Prevention of Infectious Diseases, Nagasaki University, Nagasaki 852-8523, Japan; nanboa@nagasaki-u.ac.jp; Tel.: +81-95-819-7970

**Keywords:** Epstein–Barr virus, virus maturation, viral egress, viral lytic cycle

## Abstract

The three subfamilies of herpesviruses (alphaherpesviruses, betaherpesviruses, and gammaherpesviruses) appear to share a unique mechanism for the maturation and egress of virions, mediated by several budding and fusion processes of various organelle membranes during replication, which prevents cellular membrane disruption. Newly synthesized viral DNA is packaged into capsids within the nucleus, which are subsequently released into the cytoplasm via sequential fusion (primary envelopment) and budding through the inner and outer nuclear membranes. Maturation concludes with tegumentation and the secondary envelopment of nucleocapsids, which are mediated by budding into various cell organelles. Intracellular compartments containing mature virions are transported to the plasma membrane via host vesicular trafficking machinery, where they fuse with the plasma membrane to extracellularly release mature virions. The entire process of viral maturation is orchestrated by sequential interactions between viral proteins and intracellular membranes. Compared with other herpesvirus subfamilies, the mechanisms of gammaherpesvirus maturation and egress remain poorly understood. This review summarizes the major findings, including recently updated information of the molecular mechanism underlying the maturation and egress process of the Epstein–Barr virus, a ubiquitous human gammaherpesvirus subfamily member that infects most of the population worldwide and is associated with a number of human malignancies.

## 1. Introduction

The Epstein–Barr virus (EBV) is a ubiquitous oncovirus that establishes a lifelong persistent, mostly asymptomatic, infection in more than 90% of the adult population globally. EBV can transform resting primary B cells into immortalized lymphoblastoid cells in vitro. Latent EBV infection is known to be associated with multiple lymphoid malignancies and epithelial cancers, such as Burkitt’s lymphoma, Hodgkin’s disease, EBV-associated gastric carcinoma, and nasopharyngeal carcinoma [1]. Emerging reports suggest that EBV is a major risk factor for multiple sclerosis [2]. EBV has a 170–175 kb linear double-stranded DNA genome, which is flanked on both ends by tandem terminal repeats (TRs). The EBV genome encodes approximately 90 open reading frames [1]. The genome is packaged within an icosahedral capsid, approximately 100–120 nm in diameter. Like other herpesvirus subfamilies, EBV nucleocapsids are surrounded by a protein layer that lacks a polyhedral architecture, a tegument, and an envelope [1].

EBV generally undergoes two distinct phases in its life cycle: latency and lytic infection. Viral genome replication occurs during both phases in the nucleus. Only a few viral genes are expressed during latency. The viral genome is episomally maintained by tethering to host chromosomes via the EBV nuclear antigen (EBNA) 1, where it undergoes licensed replication under the control of host replication machinery [3,4].

Spontaneous or exogenous induction initiates lytic infection, which leads to several hundred-fold amplifications of the viral DNA within 1–2 days [5]. Lytic DNA replication is regulated by viral replication machinery and produces long concatemers that are eventually cleaved at the TR and packaged as linear genome units in capsids to form nucleocapsids [6]. Nucleocapsids are subsequently enveloped and mature into infectious virions.

Herpesviruses share a similar mechanism for the maturation and egress of virions that is mediated by several budding and fusion events of various intracellular membranes during replication, which prevents membrane disruption. The synthesized viral DNA is used as a template for gene transcription, leading to translation into viral structural proteins that form capsids. Nucleocapsids initially acquire an envelope by budding through the inner nuclear membrane (INM) into the perinuclear space. This is known as the primary envelopment process. Enveloped viral particles subsequently undergo de-envelopment, which is mediated by fusion of the primary envelope with the outer nuclear membrane (ONM). This process allows nucleocapsids to release into the cytoplasm, where they acquire a layer of tegument proteins. The tegument is a hallmark of all herpesviruses and ultimately fills the space between the nucleocapsids and envelopes.

Tegument proteins play crucial roles in herpesvirus life cycle and pathogenicity, such as in the establishment of primary infection, virion maturation, and immune evasion [7]. Tegumented nucleocapsids then undergo secondary envelopment by budding into various cytoplasmic organelles, which lead to the production of mature virions in these compartments. This process is defined as secondary or final envelopment. Intracellular compartments containing mature virions are subsequently transported to the plasma membrane (PM) via the host vesicle trafficking pathway, where they fuse with the PM to release virions extracellularly.

The molecular mechanisms underlying the maturation and egress processes of herpesviruses have been the most extensively studied alphaherpesvirus subfamilies [8,9,10,11,12,13,14,15]. However, the mechanisms of gammaherpesvirus maturation and egress are poorly understood because of the lack of an efficient viral replication system. This review encompasses the current knowledge on the molecular mechanisms of EBV maturation and egress by focusing on the interaction of viral proteins with the cellular machinery.

## 2. EBV Nucleocapsid Formation under the Lytic Cycle

Latently infected cells undergo a lytic cycle upon stimulation by phorbol ester, transforming growth factor beta, histone deacetylase inhibitors, or cross-linking of immunoglobulins on the surface of B cells [16,17,18,19,20,21]. Viral lytic genes are expressed in sequential order and are divided into the following three stages: immediate-early (IE), early (E), and late (L) (Table 1). The two IE genes, origins of lytic replication (oriLyt)-binding protein *BZLF1* and immediate early transactivator *BRLF1*, are initially transcribed and coordinately bind to the origin of lytic replication (oriLyt) to induce the expression of E-genes [2]. The IE and E genes promote viral DNA synthesis via the viral DNA polymerase complex (comprising the DNA polymerase BALF5, DNA polymerase processivity factor BMRF1, and single-stranded DNA-binding protein BALF2), helicase/primase complex (comprising the helicase BBLF4, primase BSLF1, and helicase–primase-associated protein BBLF2/3), and BZLF1. These complexes accumulate in specialized subnuclear regions, defined as replication compartments, where the viral DNA is synthesized [22,23]. In addition, the viral uracil DNA glycosylase (BKRF3) contributes to viral genome replication through physical interactions with the viral DNA replication complex [24,25]. After viral DNA replication, six E gene products, BcRF1, BDLF3.5, BGLF3, BFRF2, BDLF4, and BVRF1, form a viral pre-initiation complex and activate the transcription of L genes from the newly synthesized viral genome [26,27]. L products mainly consist of structural proteins, including viral capsid proteins and glycoproteins. 

De novo synthesized viral proteins involved in EBV DNA replication and nucleocapsid formation translocate into the nucleus. In general, the active transport of proteins into the nucleus is mediated by specific nuclear localization signals (NLSs). The importin α/β heterodimer interacts with the NLSs and facilitates protein translocation across the nuclear envelope by targeting the protein to the nuclear pores. The translocation of various viral proteins into the nucleus is mediated by noncanonical NLS pathways. For example, EBV uracil DNA glycosylase BKRF3 lacks a canonical NLS, and its translocation into the nucleus is mediated by its interaction with BMRF1 [24]. Moreover, the viral cyclin-dependent kinase (CDK) 1-like protein kinase, BGLF4, is transported into the nucleus via its direct interaction with phenylalanine-glycine nucleoporins in an importin-independent manner [28]. BGLF4 also promotes the nuclear import of other DNA replication-related viral proteins that lack canonical NLSs, such as BSLF1, BBLF4, and viral capsid protein BcLF1 [28]. The replicated viral dsDNA is packaged into assembled capsids to form nucleocapsids [30]. Two tegument proteins, BGLF1 and BVRF1, are involved in this process. They further form a capsid-associated tegument complex with BPLF1 and associate with nucleocapsids [32]. A study using electron microscopy revealed that nucleocapsids are formed in the nucleoplasm near the nuclear envelope (Figure 1) [33].

## 3. Nuclear Egress of EBV Nucleocapsids

### 3.1. Budding of Primary Enveloped Nucleocapsids from the INM

Icosahedral herpesvirus nucleocapsids 100–120 nm in diameter are too large to move through the nuclear pores (central channel diameter of ~38 nm) [45]. Thus, they escape from the nucleus by vesicular transport across the nuclear envelope. The molecular mechanism underlying this process is well studied in the alpha- and betaherpesvirus subfamilies. To allow for access of viral capsids to the INM, the viral kinase pUS3 and the cellular kinase PKC cooperatively phosphorylate nuclear laminas, which locally disassemble the lamina meshwork [46,47]. The budding of vesicles into the perinuclear space requires a nuclear egress complex (NEC). The current understanding of the function of NECs is mainly based on studies of two homologs encoded by the alphaherpesvirus subfamily [herpes simplex virus (HSV)-1 and pseudorabies virus (PRV)], UL31, and UL34. UL31, a soluble nuclear phosphoprotein, and UL34, a single transmembrane protein, coordinately form a hexagonal membrane-bound NEC coat with a robust membrane budding ability [48]. NEC complexes derived from betaherpesviruses, UL53 and UL31, human cytomegalovirus (HCMV), and UL50 and UL34 from murine cytomegalovirus (MCMV) exhibited a hexagonal lattice with geometry and dimensions identical to those formed by the alphaherpesvirus NEC, which is essential for successful viral replication [49,50].

EBV-encoded BFRF1 and BFLF2, highly conserved UL34 and UL31 homologs, respectively, are involved in the nuclear egress of EBV. The overexpression of BFRF1 and BFLF2 produces curved multilayered cisternae in the INM [51]. The crystal structure of the EBV NEC exhibits five independent, structurally distinct heterodimers in the asymmetric unit [34] that appear different from the hexagonal morphology observed in the alphaherpesvirus NEC [35,36,37]. ORF67 and ORF69, NEC homologs derived from another human gammaherpesvirus subfamily member, Kaposi’s sarcoma-associated herpesvirus (KSHV), produce curved multilayered cisternae at perinuclear vesicles that resemble the structures observed in the alphaherpesvirus NEC structure [38]. These studies suggest that EBV NEC employs a unique nuclear egress mechanism owing to its structural flexibility and ability to form coats with different geometries. 

In addition, several host proteins that are responsible for primary envelopment have been identified. WD repeat-containing protein 5 (WDR5) is responsible for the nuclear egress of HCMV [39]. During EBV infection, the host endosomal sorting complex required for transport (ESCRT) machinery participates in the scission process of the INM. BFRF1 recruits the ESCRT adaptor protein Alix to the nuclear periphery. Inhibition of the ESCRT pathway inhibits BFRF1-induced vesicle formation and leads to the accumulation of both viral DNA and capsid proteins in the nucleus [51,52]. Furthermore, the ubiquitination of BFRF1 is required for vesicle formation, which is likely mediated by an unknown BFRF1-associated host ubiquitin ligase [40].

BGLF4 was also found to facilitate the nuclear egress of nucleocapsids, as well as the translocation of viral late proteins via phosphorylation and reorganization of the nuclear lamina and NPC [28,53].

### 3.2. De-Envelopment of the EBV Nucleocapsid at the ONM

The molecular mechanism underlying the de-envelopment of herpesvirus nucleocapsids remains to be elucidated because of the difficulties in capturing this process. Nonetheless, several viral and cellular proteins have been identified as regulators of HSV-1 de-envelopment. Viral glycoproteins appear to be incorporated into the primary envelope and play roles in mediating fusion between the viral primary envelope and the ONM. The lack of both gB and gH of HSV-1 was found to suppress the de-envelopment process [31]. Phosphorylation of pUL31 and gB by the viral kinase pUS3 has also been shown to be critical for the fusion process [54,55]. Moreover, several host factors such as p32, CD98 heavy chain, and β1 integrin are recruited to the nuclear membrane in HSV-1-infected cells. The downregulation or modification of these proteins results in the accumulation of primary enveloped virions in the perinuclear space or leads to INM-derived vesicles invaginating into the nucleoplasm [56,57]. Another study demonstrated that the expression of dominant negative forms of the components of the linker of the nucleoskeleton and cytoskeleton (LINC) complex resulted in the accumulation of primary enveloped HSV-1 in the perinuclear space and escape into the ER, indicating that the intact LINC complex appears to promote the fusion of the viral primary envelope with the ONM [58].

## 4. Final Envelopment of the EBV Nucleocapsid

All three herpesvirus subfamilies acquire their final envelopes in various cytoplasmic compartments, such as the cis-Golgi and trans-Golgi network (TGN) and endosomes, prior to secretion into the extracellular milieu. For alphaherpesviruses, including HSV-1, PRV, and varicella zoster virus (VZV), the final envelopment occurs in cell compartments containing the TGN [8,9,10,11,12,13,14,15] and early endosome (EE) markers [59].

Unlike alphaherpesviruses, betaherpesviruses such as HCMV and human herpesvirus type 6 (HHV-6) are reported to generate unique compartments by reorganizing pre-existing intracellular compartments containing a variety of cell organelle markers, such as the TGN, EEs, multivesicular bodies, and late endosomes (LEs) [60,61,62,63,64,65].

Electron microscopic analysis has revealed the ultrastructural basis of the various replication stages of EBV [66,67,68,69], KSHV [70], and murine gammaherpesvirus 68 (MHV-68) [71]. These studies demonstrate that tegumented capsids bud into vesicles located adjacent to the Golgi apparatus to acquire a secondary envelope. Moreover, immunofluorescence staining revealed that EBV targeted compartments containing cis-Golgi and TGN markers to gain the final envelope, followed by subsequent transport to the PM for the release of matured virions [33] (Figure 1). These findings suggest that EBV shares its mechanisms of virion maturation and release with other herpesvirus subfamilies. In addition, a herpesvirus infection can induce autophagy, and the autophagic membrane is reported to be used by an alphaherpesvirus subfamily member, VZV, for secondary envelopment [15,72,73]. Another study demonstrated that macroautophagic membranes are stabilized in the cells that undergo the lytic cycle of EBV, and macroautophagy-related proteins are incorporated into EBV particles [73]. These studies indicate that EBV exploits various cellular machineries for the efficient acquisition of the secondary envelope.

Mature viral membrane glycoproteins are incorporated in the intercellular compartments for final envelopment. Glycoproteins contain targeting signals that are consistent with the evidence that the TGN that plays a critical role in viral final envelopment. These glycoproteins containing oligosaccharide chains covalently attach to polypeptides and are guided to the TGN by signal peptides. The sorting sequence of glycoproteins is also important for their assembly into virions. The recruitment of gE of VZV to the TGN depends on an AYRV motif and an acidic amino acid–rich domain in the cytoplasm [8]. Similarly, the cytoplasmic domain of gE of HSV is responsible for its accumulation in the TGN in the early stage of infection [74]. gBs of HSV and PRV also have signal peptides in the cytoplasmic domain that guide them to the TGN [75].

Tegument proteins contain sorting signals targeting specific intracellular compartments where the viral glycoproteins are loaded. The interaction between tegument proteins and viral glycoproteins in the compartments is essential for recruiting capsids to the sites of final envelopment [76,77]. HSV UL11 and UL16, which are conserved in all herpesvirus subfamilies, coordinately bind to gE; this process contributes to the acquisition of the final envelope and release of virions [78]. A recombinant HSV lacking gE-gI generates large aggregates of unenveloped capsids in the cytoplasm [31]. A lack of MHV-68-encoded ORF33, which is the homolog of HSV UL16, also showed a similar phenotype [30]. These reports demonstrate the critical roles of tegument proteins in promoting the final envelopment, although the host factors involved in this process remain to be elucidated.

The purified EBV particles contained at least 17 tegument proteins [79]. Several tegument proteins are involved in viral maturation. Comprehensive analysis of the intraviral protein interactome has shown that BLRF2 is vital for efficient tegumentation [26]. Another study demonstrated that the viral DNA-binding protein BALF2 exploits the small GTPase Rab1, which is involved in vesicle trafficking between the endoplasmic reticulum (ER) and Golgi to target BALF2 itself in the viral assembly compartment and contribute to the proper glycosylation of the major viral glycoprotein gp350/220 [80]. Another study demonstrated that the EBV tegument protein BBLF1, a homolog of UL11 for HSV-1 and UL99 for HCMV, is involved in the final envelopment [44]. Both endogenous and exogenously expressed BBLF1 predominantly co-localized with a TGN marker; by contrast, exogenously expressed BBLF1 partially co-localized with a cis-Golgi marker [44]. A protein interactome analysis revealed that the capsid-associated BGLF2 interacts with the tegument protein BBLF1, which likely facilitates the interaction of capsids with viral glycoprotein-loaded Golgi membranes during final envelopment [80]. The major EBV tegument protein BNRF1, which is encoded only by gammaherpesvirus subfamily members, distributes in both the cytoplasm and nucleus [41]. BNRF1 localizes to the promyelocytic leukemia (PML) nuclear bodies; however, the significance of its specific distribution has not been characterized in detail. Another tegument protein, BRRF2, which is conserved only in KSHV and MHV-68, is involved in viral production [43]. Exogenously expressed BRRF2 is partly localized with an EE marker but not with ER and Golgi apparatus markers. Inconsistency in the intracellular distribution of these tegument proteins may reflect the overexpression of the protein; therefore, further investigation of individual tegument proteins is required in virally infected cells that undergo the lytic cycle.

## 5. Trafficking of Matured Virions to the Plasma Membrane and Extracellular Release

Ultimately, all herpesviruses are released extracellularly by the fusion of the intracellular compartment containing mature virions with the PM. Effective viral release depends on host membrane traffic mechanisms that allow for mature virions to be transported to the cell surface, as well as these final fusion events.

The secretory pathway plays a role in trafficking newly synthesized proteins to the PM, where they can be released. These proteins are transported to the Golgi complex for further processing and sorting. Proteins are stored in the secretory vesicles, which are formed from the TGN and are further post-translationally modified. Upon various stimuli, secretory vesicles traffic to the cell surface and fuse with the PM via exocytosis, which allows the proteins to be rapidly secreted. The small GTPase Rab family regulates the biogenesis, transport, tethering, and fusion of cell organelles and vesicles. This GTPase activity is regulated by guanine nucleotide exchange factors (GEFs) and GTPase-activating proteins (GAPs), which allow Rabs to cycle between an active (GTP-loaded) and an inactive (GDP-loaded) state. GTP-loaded Rabs localize to specific membranes of several compartments, such as the ER, Golgi apparatus, secretory vesicles, and endosomes, where they recruit effector proteins that regulate different steps of membrane trafficking [29]. Secretory vesicle traffic is thought to be regulated by specific Rab family members such as Rab6, Rab8, Rab10, and Rab11.

Several lines of evidence have demonstrated that herpesvirus subfamilies exploit host secretory pathways for mature virion trafficking and release into the extracellular milieu. Some studies have characterized the mechanisms of the release of alphaherpesviruses. Live-cell imaging with a fluorescence microscopy revealed that vesicles containing PRV were associated with Rab6a, Rab8a, and Rab11a, which are involved in secretory pathway-mediated vesicular transport in a microtubule-dependent manner [11]. Additionally, other studies have demonstrated that the motor protein myosin Va, which normally plays a role in the transport of secretory granules to the PM, is activated in HSV-1-infected cells. The same study indicated that this molecule is involved in the cortical actin-mediated transport of virions from the TGN to the PM [81].

Regarding the egress process for EBV, immunofluorescence staining revealed that small GTPases involved in host secretory machinery, including Rab8a, Rab10, and Rab11a, partially co-localized with a EBV glycoprotein, gp350/220. Furthermore, the downregulation of these molecules led to the accumulation of viral structural proteins in the cytoplasm by inhibiting their traffic to the PM and subsequent release of EBV infectious virions. These results suggest that mature EBV particles are released into the extracellular milieu via the host secretory pathway [82] (Figure 1). Only a few viral and host genes involved in viral egress have been identified. BBRF3 (gN/gM) [68] and the KSHV glycoprotein gB [83] contribute to viral assembly and egress. Another study, using a combination of electron microscopy and subcellular fractionation analyses, demonstrated that BBLF1 knockout leads to the accumulation of virions in secretory vesicles, suggesting its importance in EBV virion release [42]. A recent study demonstrated that BGLF4 induces cytoskeletal rearrangement, which is the coordinately induced redistribution of cytoplasmic organelles. This study also identified a scaffold protein involved in regulating the dynamics and assembly of the actin cytoskeleton IQ motif–containing GTPase-activating protein 1 (IQGAP1) as a responsible host factor for virus maturation and subsequent virion release [84].

## 6. Conclusions and Perspectives

All herpesvirus subfamilies replicate viral DNAs and form nucleocapsids in the nucleus; therefore, they must achieve two independent envelopment processes during viral maturation. Herpesviruses commonly transfer nucleocapsids into the cytoplasm by budding into the INM and fusing with the ONM. This strategy prevents the disruption of the nuclear envelope. Once in the cytoplasm, herpesviruses bud into cytoplasmic compartments, similar to other enveloped viruses that replicate and assemble in the cytoplasm.

Although recent studies have uncovered the mechanism underlying the maturation and egress of EBV particles, the detailed molecular mechanism remains unclear. In particular, the role of host counterparts of tegument proteins in the process of virion assembly remains to be fully understood. A further characterization of individual tegument proteins is essential in the whole process of EBV maturation. It is particularly important to better understand how nucleocapsids are preferentially recognized for envelopment, and whether and how viral glycoproteins participate in both primary and final envelopment. Finally, the assembly of the tegument layer and the interaction of tegument proteins with the cell organelle membrane or glycoproteins guide secondary envelopment.

Several findings suggest the importance of lytic infection in the development of EBV-associated cancers [85]. Therefore, further investigation of the molecular mechanism underlying virion maturation will provide insights into EBV-mediated oncogenesis.

## Figures and Tables

**Figure 1 microorganisms-12-00806-f001:**
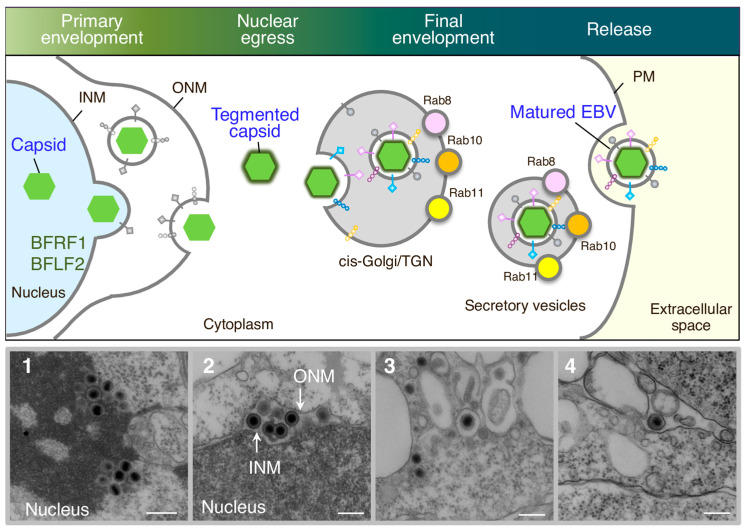
Overview of EBV maturation. Newly synthesized viral DNAs are packaged into capsids in the nucleoplasm. Nucleocapsids acquire primary envelopes by budding through the inner nuclear membrane (INM) into the perinuclear space (primary envelopment). The primary enveloped virus then undergoes de-envelopment, which is mediated by the fusion of the primary envelope with the outer nuclear membrane (ONM) (nuclear egress). This process is mediated by viral nuclear egress complexes such as BFRF1 and BFLF2. The released nucleocapsids in the cytoplasm then acquire a coat of tegument proteins. Tegumented nucleocapsids then undergo a final envelopment by budding into intracellular compartments containing the cis-Golgi and trans-Golgi networks (TGN) markers to produce mature virions. Compartments containing mature virions are transported to the plasma membrane (PM) via secretory machinery, which is mediated by Rab8a, Rab10, and Rab11a. The fusion of compartments with the PM allow matured EBV to release into the extracellular milieu. Electron-micrograph-visualized formation of nucleocapsids in the nucleus (1), released primary enveloped nucleocapsids into the perinuclear space (2), released nucleocapsids into the cytoplasm and secondary enveloped EBV in the intracellular compartments (3), and extracellularly released mature EBV (4). Scale bars: 250 nm.

**Table 1 microorganisms-12-00806-t001:** Summary of Responsible EBV Genes for Virion Maturation.

Herpesvirus Homologues				Function in EBV Life Cycle
EBV	KSHV	HSV	HCMV	
EBNA1	LANA1			Maintenance of viral genome, binding to latent replication origin [4,5]
BZLF1	K08			Trans-activator, oriLyt-binding protein [3]
BRLF1	ORF50			Transcriptional activator [3]
BALF5	ORF9	UL30	UL54	DNA polymerase [23,24]
BMRF1	ORF59	UL42	UL44	DNA polymerase processivity factor, part of DNA polymerase complex [23,24]
BALF2	ORF6	UL29	UL57	Single-stranded DNA-binding protein, part of DNA polymerase complex [23,24]
BBLF4	ORF44	UL5	UL105	Helicase, part of helicase/primase complex [23,24]
BSLF1	ORF56	UL52	UL70	Primase, part of helicase/primase complex [23,24]
BBLF2/3	ORF40/41	UL8	UL203	Helicase-primase-associated protein, part of helicase/primase complex [23,24]
BKRF3	ORF46	UL2	UL114	Uracil DNA glycosylase [25,26]
BcRF1	ORF24	UL87	UL87	Interleukin-10 homologue, part of viral pre-initiation complex [27,28]
BDLF3.5	ORF35	UL14	UL96	Unknown, part of viral pre-initiation complex [27,28,29]
BGLF3	ORF34	UL34	UL95	Part of viral pre-initiation complex [27,28]
BFRF2	ORF66	UL31	UL31	Part of viral pre-initiation complex [27,28]
BDLF4	ORF31	U63 *	UL92	Part of viral pre-initiation complex [27,28]
BVRF1	ORF19	UL25	UL77	Part of viral pre-initiation complex [27,28]
BGLF4	ORF36	UL13	U97	CDK-like protein kinase [30], late gene expression, initial envelopment [30,31]
BcLF1		UL19	UL86	Major viral capsid protein
BGLF1	ORF32	UL17	UL93	Potential tegument protein, Nucleocapsid formation [32]
BVRF1	ORF19	UL25	UL77	Potential tegument protein, Nucleocapsid formation [32]
BPLF1	ORF64	UL36	UL48	Nucleocapsid formation [32,33]
BFRF1	ORF67	UL34	UL53	Part of nuclear egress complex [34,35,36,37,38,39,40]
BFLF2	ORF69	UL53	UL50	Part of nuclear egress complex [35,36,37,38,39]
BBLF1	ORF38	UL11	UL99	Egress [41,42], relocalize BGLF2 [41]
BNRF1	ORF75			Major EBV tegument protein [43]
BRRF2	ORF48			Increase progeny production [29]
BGLF2	ORF33	UL16	UL94	Viral egress [29], interact with BBLF1 [44]

* encoded by HSV-6.

## Data Availability

No new data were created or analyzed in this study. Data sharing is not applicable to this article.

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
