# Peer review of "Current Insights into the Maturation of Epstein–Barr Virus Particles"

_microorganisms, 2024, doi:10.3390/microorganisms12040806_

Round 1

Reviewer 1 Report

Comments and Suggestions for Authors

This is an excellent review article on the maturation of the EBV virus. It is written in a clear scientific language and presents well-organized, established data. The article provides a comprehensive overview of the available literature, and the selection of articles is appropriate.

Author Response

I highly appreciate reviewer 1’s comments.

Reviewer 2 Report

Comments and Suggestions for Authors

1.     The manuscript has 41% similarity by iThenticate report, so the authors have to rewrite and rephrase the manuscript throughly and completely.

2.     This review summarizes the major findings, including recently updated information of the molecular mechanism underlying the maturation and egress process of Epstein–Barr virus (EBV) that is associated with a number of human malignancies. Actually, EBV is an oncovirus that may be associated with cancers, such as Nasopharyngeal carcinoma, Burkitt's lymphoma and Hodgkin's lymphoma. Please add a section to review the correlation between the maturation of EBV particles and cancers.

3.     The EBV has a charateristic to infect latently in B cells and epithelial cells. Please add a section to review the correlation between the maturation of EBV particles and latency.

Comments on the Quality of English Language

No.

Author Response

1. The manuscript has 41% similarity by iThenticate report, so the authors have to rewrite and rephrase the manuscript throughly and completely.

Thank you for your suggestion. Duplication check for this manuscript has been performed by the journal editorial office before peer review process. This manuscript has been also checked by a professional language editing service before submission.

2. This review summarizes the major findings, including recently updated information of the molecular mechanism underlying the maturation and egress process of Epstein–Barr virus (EBV) that is associated with a number of human malignancies. Actually, EBV is an oncovirus that may be associated with cancers, such as Nasopharyngeal carcinoma, Burkitt's lymphoma and Hodgkin's lymphoma. Please add a section to review the correlation between the maturation of EBV particles and cancers.

Thank you for valuable comment. In response to the suggestion, I added the following sentences in the Conclusions and Perspectives along with the relevant review. “Several findings suggest the importance of lytic infection in the development of EBV-associated cancers [85]. Therefore, further investigation of the molecular mechanism underlying virion maturation will provide insights into EBV-mediated oncogenesis (page 8, lines 319-321)."

3. The EBV has a charateristic to infect latently in B cells and epithelial cells. Please add a section to review the correlation between the maturation of EBV particles and latency.

I appreciate reviewer 2's comment. In latency only limited viral genes are expressed, which results in lack of production of infectious virions. I have modified the text as follows: Lytic DNA replication is regulated by viral replication machinery and produces long concatemers that are eventually cleaved at the TR and packaged as linear genome units in capsids to form nucleocapsids [6]. Nucleocapsids are subsequently enveloped and mature into infectious virions (page 2, lines 47-51).

Round 2

Reviewer 2 Report

Comments and Suggestions for Authors

The manuscript has been significantly improved. 

Comments on the Quality of English Language

The manuscript has to be checked similarity once more before publication.